# Single-cell sequencing reveals the response mechanisms of vascular endothelial cells to glucocorticoids in diabetic retinopathy

**Lingda Wang, Rong Zhang, Lin Wang, Yongrui Wang, Xiaodan Zhang, Guohong Zhou** ◉ *

Department of Ophthalmology, Shanxi Eye Hospital Affiliated to Shanxi Medical University, Taiyuan, Shanxi, China

* guohongzhou2016@163.com

## Abstract

### Background

One serious consequence of diabetes mellitus is diabetic retinopathy (DR), which impairs eyesight to the point of blindness. While glucocorticoid medications are commonly employed in the management of DR, their therapeutic efficacy requires enhancement. Due to the tight association between glucocorticoid-related genes and the onset and development of DR, a comprehensive examination of its root cause of activity may be able to overcome the drawbacks of existing treatment approaches.

### Methods

R programming tools were used to examine the single-cell RNA sequencing (scRNA-seq) dataset GSE178121, which was obtained from the Gene Expression Omnibus (GEO) database. To evaluate glucocorticoid activity, a gene set related to glucocorticoid phenotypes was sourced from the Molecular Signatures Database (MSigDB), followed by the identification of key cellular populations within DR tissues. Subsequently, these key cells underwent pseudotime analysis, transcription factor (TF) evaluation, cell-cell communication assessment, differential gene screening, and the construction of a regulatory network.

### Results

Our investigation demonstrated that vascular endothelial cells (VECs) in DR tissue exhibited markedly elevated glucocorticoid activity. KLF4 is among the TFs that are intimately linked to the onset of DR, and hydroxyurea could be a beneficial medication. Cell-cell communication analysis highlighted the PTN and ANGPTL signaling pathways as important signaling pathways in DR. In the meanwhile, we identified 25 Hub genes, including *DUSP6*, *AP1S2*, and *PTPRB*, which were verified to be differentially expressed in DR.

**Data availability statement:** All relevant data are within the paper and its Supporting information files.

**Funding:** This study was supported by the Shanxi Provincial Basic Research Program (Free Exploration Category) awarded to G. Z. (No. 202403021221330), the Shanxi Provincial Optimal Funding Program for Scientific and Technological Activities of Returned Overseas Scholars awarded to G. Z. (No. 20240057), the Scientific Research Project of Shanxi Provincial Health Commission awarded to G. Z. (No. 2023042), and the In-hospital Scientific Research Fund of Shanxi Eye Hospital awarded to G. Z. (No. C201801). The specific roles of this author are articulated in the "author contributions" section. The funders had no role in study design, data collection and analysis, decision to publish, or preparation of the manuscript.

**Competing interests:** The authors have declared that no competing interests exist.

## Conclusion

In conclusion, our comprehensive study elucidated the complex interactions of glucocorticoids in the pathogenesis of DR, thereby revealing potential signaling pathways and therapeutic targets.

---

## Introduction

Diabetic retinopathy (DR), a severe microvascular consequence of diabetes mellitus, is one of the primary causes of adult blindness globally. Nearly one-third of people with diabetes are predicted to develop some form of DR, and 10% may develop vision-threatening diseases such diabetic macular edema (DME) or proliferative diabetic retinopathy (PDR) [1]. The incidence of DR is anticipated to escalate alongside the global diabetes epidemic, which is projected to affect 700 million individuals by the year 2045 [2]. Treatment effectiveness is still below ideal despite improvements in therapy choices, such as intravitreal injections of glucocorticoids, anti-vascular endothelial growth factor (VEGF) medicines, and laser photocoagulation [3]. Therefore, more in-depth pathogenic mechanisms and more effective therapeutic targets need to be explored.

Glucocorticoids are steroid hormones that regulate various physiological processes, including metabolism, immune response, and inflammation. The dysregulation of glucocorticoid signaling has been associated with numerous chronic inflammatory diseases, including DR [4]. Currently, the main applications of glucocorticoids in DR include intravitreal injections of triamcinolone acetonide (TA), Ozurdex and Iluvien [5]. However, intravitreal injections of glucocorticoids are often less effective than anti-VEGF therapy [6].

Single-cell RNA sequencing (scRNA-seq) is a landmark technology that can help scientists explore the cellular and molecular basis of complex diseases [7]. Finding novel therapeutic targets and identifying gene expression patterns unique to cell types are made possible by this cutting-edge technique. Previous studies have demonstrated the effectiveness of scRNA-seq in identifying genetic markers and cellular heterogeneity across a range of eye disorders. For instance, scRNA-seq analysis of retinal cell gene expression has illuminated distinct patterns linked to glaucoma and age-related macular degeneration (AMD) [8,9]. Additionally, scRNA-seq has been instrumental in discovering novel biomarkers and therapeutic targets for DR, including specific microRNAs and cytokines that play pivotal roles in retinal inflammation and angiogenesis [10]. These findings underscore the potential of scRNA-seq in advancing our understanding of DR pathogenesis and identifying new therapeutic avenues.

In this work, we investigated the role of glucocorticoid-related genes in the pathophysiology of DR using scRNA-seq. We extracted single-cell transcriptomic data from retinal tissues of both DR and control mice and conducted thorough bioinformatics analyses to pinpoint differentially expressed genes (DEGs) and associated pathways. Our primary focus was on glucocorticoid-related genes and their regulatory networks, which encompass transcription factors (TFs), non-coding RNAs, and protein-protein interactions. In order to shed light on the biological pathways and processes connected to these genes,

we also conducted functional enrichment studies. Our results revealed significant alterations in glucocorticoid-related gene expression in DR retinal cells, suggesting their involvement in disease pathogenesis. We identified key TFs and non-coding RNAs that may regulate glucocorticoid signaling in DR. Additionally, our functional enrichment analyses revealed several biological processes and pathways, including inflammation, angiogenesis, and extracellular matrix remodeling, which may be influenced by glucocorticoid-related genes in the context of DR.

Overall, this study offers novel perspectives on the molecular mechanisms that underlie DR and pinpoints potential therapeutic targets for this debilitating condition. The findings provide support for the creation of focused therapies meant to lessen vision loss in diabetic patients and highlight the crucial part that glucocorticoid signaling plays in the pathophysiology of DR. The medication intervention, pathway crosstalk, and related biomolecules discussed here may offer fresh perspectives on enhancing the therapeutic impact of glucocorticoids in DR.

## Materials and methods

### Single-cell data downloading and processing

The Gene Expression Omnibus (GEO) database houses extensive single-cell sequencing datasets, and in the study, the scRNA-seq dataset GSE178121 of mouse DR tissues, which was sequenced based on the GPL24247 Illumina NovaSeq 6000 (Mus musculus) platform sequencing, which contains a total of 6 samples, including 3 DR tissue sample and 3 control sample [11]. Initially, low-quality cells and genes were excluded based on the following criteria: 1) Genes with undetectable expression were eliminated. 2) Cells exhibiting a variable number of expressed genes within the range of 200–8000 were retained. 3) Cells with fewer than 30000 unique molecular identifiers (UMIs) were maintained. 4) Cells with a mitochondrial gene percentage below 20% were included. The Seurat R software package's "normalizedata" function was used to normalize the data. After normalization, we used an algorithm that balanced variance and average expression to find highly variable genes. Principal component analysis (PCA) was subsequently conducted, with significant principal components (PCs) employed as input for graph-based clustering. The HARMONY technique was used to reduce batch effects across samples. We used the "FindClusters" tool for clustering, which uses the Shared Nearest Neighbor (SNN) modular optimization approach, resulting in 20 clusters across 20 PC components with a resolution of 0.5. After that, the "Runtsne" function was used to t-Distributed Stochastic Neighbor Embedding (t-SNE), where t-SNE-1 and t-SNE-2 were used to display cell clustering. To identify DEGs among the cell clusters, the "FindAllMarkers" function was applied to the normalized gene expression data using default parameters set by Seurat. Following this, cells were identified through cell type-specific biomarkers, and their respective ratios were calculated and evaluated.

### Glucocorticoid-related gene downloads and scores

Glucocorticoid-related gene data were obtained from the MSigDB database (https://www.gsea-msigdb.org). R package "AUCell" was utilized to score each cell based on gene set enrichment analysis [12]. Based on the area under the curve (AUC) values of the 363 selected glucocorticoid-related genes (S1 Table), the gene expression ranking for each cell was generated (sorted in descending order based on the raw expression levels within each cell) to estimate the proportion of highly expressed gene sets in each cell. Cells that express more genes in the gene set have higher AUC values. The "AUCell_exploreThresholds" function was used to determine an appropriate threshold in order to detect active cells within the gene collection. Subsequently, the AUC scores for each cell were visualized in t-SNE embeddings via the "ggplot2" R package, facilitating the observation of clusters of activated cells.

### Analysis of cell differences between groups

Differential expression analysis between the identified key cells of the DR group and the Control group was conducted at the single-cell level. The "limma" R package was used to discover DEGs in those critical cells, with a selection criterion of |logFC|>0.25 and P value<0.05 for further investigation [13].

## Enrichment analysis of GO and KEGG

An investigation of Gene Ontology (GO) enrichment including biological processes (BP), molecular functions (MF), and cellular components (CC) was conducted [14]. A bioinformatics tool for locating gene lists that are substantially enriched in modified metabolic pathways is the Kyoto Encyclopedia of Genes and Genomes (KEGG) [15]. The R package "clusterPro-filer" facilitated GO and KEGG enrichment analysis, with a significance level set at P < 0.05 [16].

## Transcription factor regulatory network analysis

To find important TFs in various cell clusters, a cis-regulatory study was carried out using pySCENI, a program for deter-mining gene regulatory networks based on DNA sequence and co-expression analysis [17]. The AUC was then computed to evaluate every cell's network activity.

In this process, GENIE3 was employed to identify TFs, which were then organized into modules (rules), followed by gene-matrix sequencing analysis utilizing RcisTarget, focusing on 500 bp upstream and 100 bp downstream of the tran-scriptional start site (TSS). The rule activity of each cell was scored using AUCell, and binarized regulon activity was subsequently plotted on t-SNE representations. TFs adjusted for Benjamini-Hochberg error discovery rate <0.05 were taken into consideration for further research. Next, the correlation between rules and IFN-I scores was analyzed using the Pearson correlation coefficient.

## Construction of cell trajectories by pseudotime analysis

The pseudotime analysis was carried out using Monocle 2, which creates a pseudotime graph that may take branching and linear differentiation processes into account by performing reverse graph embedding based on a user-defined list of genes [18]. The raw count data were normalized by generating a size factor for trajectory inference in order to do the pseudotime analysis of critical cells. Cell trajectories were constructed from genes exhibiting a high degree of discreti-zation and substantial expression levels [19]. The DDRTree algorithm's default settings were used. We used Branching Expression Analysis Modeling (BEAM), which is integrated into Monocle 2, to look into branching events in more detail. BEAM finds genes that have high branching-dependent expression. Branch-dependent expression patterns were visual-ized as heatmaps using Monocle 2.

## Cell-cell communication analysis and expression of ligands and receptors

The "CellChat" R program was used to create CellChat objects based on the UMI count matrix for both the DR and Control groups in order to analyze cell-cell communication and ligand-receptor expression [20]. The "CellChatDB.human" ligand-receptor interaction database was used as a guide for the investigation of cell-cell communication. Default settings were used to create the reference database. CellChat objects from each group were integrated using the "mergeCellChat" function, which allowed for a comparison of the overall number of interactions as well as the intensity of those interac-tions. The "netVisual_diffInteraction" function was used to show the differences in the number or strength of interactions between different cell types within each group. Additionally, the distribution of signaling gene expression across the vari-ous groups was shown using the "netVisual_bubble" and "netVisual_aggregate" algorithms.

## Identification of Hub genes

Protein pairings were filtered based on interaction scores greater than 0.7 using the Search Tool for the Retrieval of Inter-acting Genes (STRING) online database (https://string-db.org) to create the Protein-Protein Interaction (PPI) network [21]. After that, the PPI network was visualized more clearly using Cytoscape software [22]. Hub genes were identified through the Cytoscape plugin, CytoHubba, which allowed for the extraction of the top 50 genes from all 12 algorithms available in CytoHubba. The intersection of these genes was designated as the hub genes for further analysis [23]. The algorithms

used in CytoHubba included Betweenness, Stress, Radiality, BottleNeck, Maximum Neighborhood Component (MNC), Maximal Clique Centrality (MCC), Degree, Density of Maximum Neighborhood Component (DMNC), Edge Percolated Component (EPC), Eccentricity, Clustering Coefficient, and Closeness.

## RBP-mRNA network construction

In this work, the interaction between mRNA and RBP (RNA-binding protein) expression has been explored through the commonly used freely accessible platform StarBase (https://starbase.sysu.edu.cn/tutorialAPI.php#R BPTarget). In the context of disease, the criteria set for identifying significant mRNA-RBP pairs included P < 0.05, clusterNum ≥ 5, and clip-ExpNum ≥ 5. Following this, RBP-mRNA networks were constructed using Cytoscape software.

## TF-mRNA network construction

By attaching to certain DNA sequences, TFs serve a crucial role in controlling transcriptional activities, which in turn affects a wide range of intricate biological processes. The hTFtarget database (http://bioinfo.life.hust.edu.cn/hTFtarget#!) aggregates comprehensive TF target regulation sourced from extensive human TF ChIP-Seq data, encompassing 7,190 experimental samples across 659 TFs and spanning 569 conditions (399 cell lines, 129 tissue types or cell classes, and 141 treatments). This enables the prediction of common TFs associated with multiple genes.

## ceRNA network construction

Given the unclear mechanisms by which competing endogenous RNAs (ceRNAs) function in DR, we employed miRTar-Base (https://mirtarbase.cuhk.edu.cn/~miRTarBase/miRTarBase_2022/php/index.php), StarBase 2.0 (https://starbase.sysu.edu.cn/starbase2/index.php), and miRDB (https://mirdb.org/index.html) databases to reverse predict key gene microRNAs, aiming to seek lncRNAs associated with common microRNAs of these hub genes, ultimately constructing ceRNA networks [24,25].

## DGIdb drug prediction

A publicly accessible resource, the Drug-Gene Interaction Database (DGIdb, https://dgidb.org) compiles information on gene or gene product interactions with medications and drug-gene interaction data to help researchers and physicians generate hypotheses [26]. We used the DGIdb database to predict potential drugs or small molecules that could combine with the noticed key genes.

## Cell culture and processing

Human retinal microvascular endothelial cells (HRMECs) were cultured in endothelial cell medium (ECM). To simulate the environments of normal conditions and DR, HRMECs in the control group and high-glucose group were treated with 5.5 mM glucose and 40 mM glucose, respectively, for 48 hours.

## Quantitative real-time polymerase chain reaction (qRT-PCR)

Total RNA was extracted from cells using the Ultrapure RNA Kit (CW0581M, CWBIO) according to the manufacturer's instructions. The concentration and purity of the extracted RNA were determined using an ultraviolet spectrophotometer (NP80, Nano-Photometer). Subsequently, the RNA was reverse-transcribed into cDNA using HiScript II Q RT SuperMix for qPCR (+gDNA wiper) (R223-01, Vazyme). qRT-PCR was performed on a real-time PCR system using SuperStar Universal SYBR Master Mix (CW3360M, CWBIO), and the results were analyzed by the $2^{-\Delta\Delta Ct}$ method. The primer sequences (5'–3') were as follows: β-actin (forward: TGGCACCCAGCACAATGAA; reverse: CTAAGTCATAGTCCGCCTAGAAGCA), DUSP6 (forward: GAACT-GTGGTGTCTTGGTACATT; reverse: GTTCATCGACAGATTGAGCTTCT), AP1S2 (forward: TTCAGACCGTTTTAGCACGGA;

reverse: TGTCCTGATCCTCAATAGCACA), and PTPRB (forward: ACAACACCACATACGGATGTAAC; reverse: CCTAGCAG-GAGGTAAAGGATCT). β-actin was used as the internal control.

## Statistical analysis

R program was used for statistical analysis of the current study. The connection between two parameters was inferred using the Spearman correlation test. While the Wilcoxon test was used to assess differences between two groups, the Kruskal-Wallis test was employed to examine differences between three or more groups. A statistically significant result was defined as a two-sided P<0.05.

Each set of qRT-PCR was repeated 3 times. All data were statistically analyzed and plotted using GraphPad Prism 10, and the study results were expressed as mean±standard deviation. Unpaired two-tailed Student's t test was used for comparison between groups. P-value<0.05 was considered statistically significant. Among them, the significance levels were indicated by the number of asterisks: *P<0.05; **P<0.01; ***P<0.001; ****P<0.0001.

## Results

Fig 1 displays the flow chart for the above study.

### Single-cell clustering and annotation with reduced dimensionality

We demonstrated the complex cellular environment in DR tissues by downscaling clustering annotation of the scRNA-seq dataset GSE178121. After initial quality control assessment and double cell removal, we successfully screened 5818 cells from single cells. Subsequently, we applied a clustering technique to divide all of the cells into 20 various classes (Fig 2A). Using singleR packages, these cell types were annotated following the gene expression patterns of each cluster

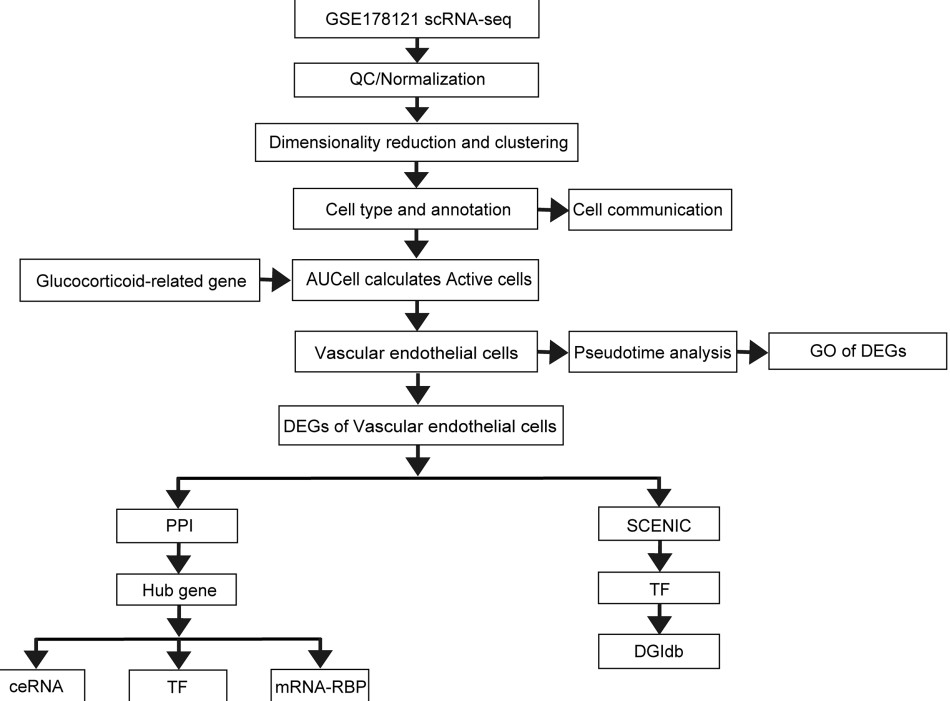

**Fig 1. Flowchart of this study.**

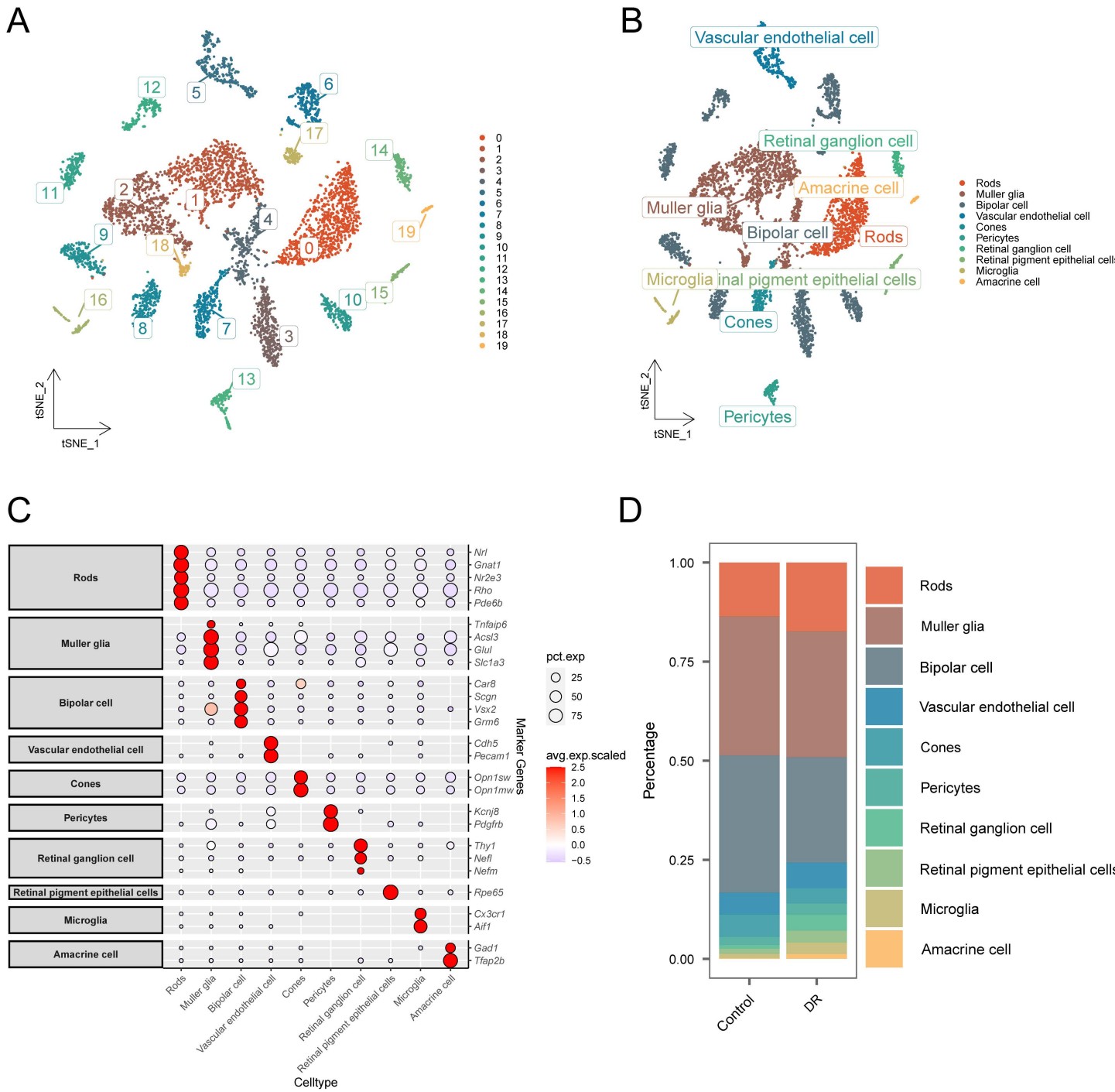

**Fig 2. Annotation and visualization of the cellular environment in DR tissues.** (A) t-SNE plot demonstrating the distribution of 20 cell subpopulations in the DR group versus the Control group. (B) The t-SNE plot demonstrates the annotation results and distribution of 10 major cell types in DR tissues. (C) The various cell type's expression of marker genes. (D) Stacked bar graph demonstrating the ratio of various cell types in the DR group versus the Control group.

in combination with certain cell biomarkers (S2 Table). We identified 10 major cell types in DR tissues, including Rods, Muller glia, and Cones (Fig 2B and 2C). Fig 2D depicts the proportional distribution of various cell types in the DR group and the Control group, revealing that there may be certain changes in different cell types before and after the onset of DR. However, alterations in cell type composition may not be a significant contributor to the pathogenesis of DR, highlighting the necessity for further investigations into the modifications in signaling interactions among cells.

## Identification of glucocorticoid activity

To determine the primary site of glucocorticoid activity in DR tissues, we examined the expression patterns of glucocorticoid-related genes to pinpoint cell clusters that exhibit the highest levels of activity at the single-cell level. 130 glucocorticoid-active cells were identified by the defined threshold, where cell populations with an area AUC value greater than 0.13 were categorized as having high glucocorticoid activity, and those with values less than 0.13 were classified as having low glucocorticoid activity (Fig 3A). We further calculated the glucocorticoid AUC scores corresponding to each cell type and compared their differences before and after the onset of DR. Fig 3B shows the t-SNE plots of the active cells, in which Vascular endothelial cells (VECs) showed significantly elevated glucocorticoid activity after the onset of DR, and thus we considered them as the main target for subsequent analysis. When we achieved a differential gene analysis for VECs comparing the DR and Control groups, we discovered 119 DEGs (S3 Table). These DEGs were then subjected to GO and KEGG enrichment analysis; the outcomes are shown in Fig 3C and 3D (S4 and S5 Tables).

## Pseudotime analysis of activated cells

We constructed a pseudotime cell trajectory based on the active cell population (VECs) to identify critical gene expression programs that influence the progression of DR. Notably, the transcriptional states along the trajectory reveal distinct biological processes. the VECs forms 1 node and splits into 2 different branches at 1 node, presenting 2 different transcriptional stages (Fig 4A–C). We examined the distribution of these various states within the DR and Control groups (Fig 4D), as well as the proportions of the DR and Control groups across these states (Fig 4E).

Based on node 1, there are two main differentiation branches of VECs in DR The genes highly expressed in pre branch (state 1) are mainly enriched in "nuclear receptor activity", "ligand-activated TF activity" and "nucleotidyltransfer activity", while the cells in cell branch 2 (state 3) are mainly enriched in "visual perception", "sensor perception of light stimulus", "response to light stimulus" and other related pathways. Genes enriched in "tRNA aminoacylation", "amino acid activation", "protein O-linked glycosylation" and "cis-Golgi network" were highly expressed in cell Branch 1 (state 2) (Fig 4F) (S6 Table).

## Analysis of transcription factors in activated cells

We explored the TFs in VECs that changed significantly between groups. The results are shown in Fig 5A. A total of 11 TFs that changed significantly between groups were identified, namely: Arid3a, Atf3, Bclaf1, Fos, Sp1, Sp2, Klf4, Rax, Sox17, Sox5, Tbp. We analyzed the expression of 11 TFs in all VECs (Fig 5B), with low expression of Rax and Sox5.

We also analyzed the distribution of the 11 TFs in all VECs, and the results are shown in Fig 5C. Except for Rax and Sox5 with low expression in VECs, the remaining 9 TFs were highly expressed in VECs.

## Small molecule drug forecasting

Eleven TFs were uploaded to the official website of DGIdb (https://www.dgidb.org) for analysis, and finally only four TFs successfully predicted small molecule drugs. In descending order of Interaction score, the small molecule drug with the highest score and regulatory approval of APPROVED status was obtained as the final prediction (Table 1).

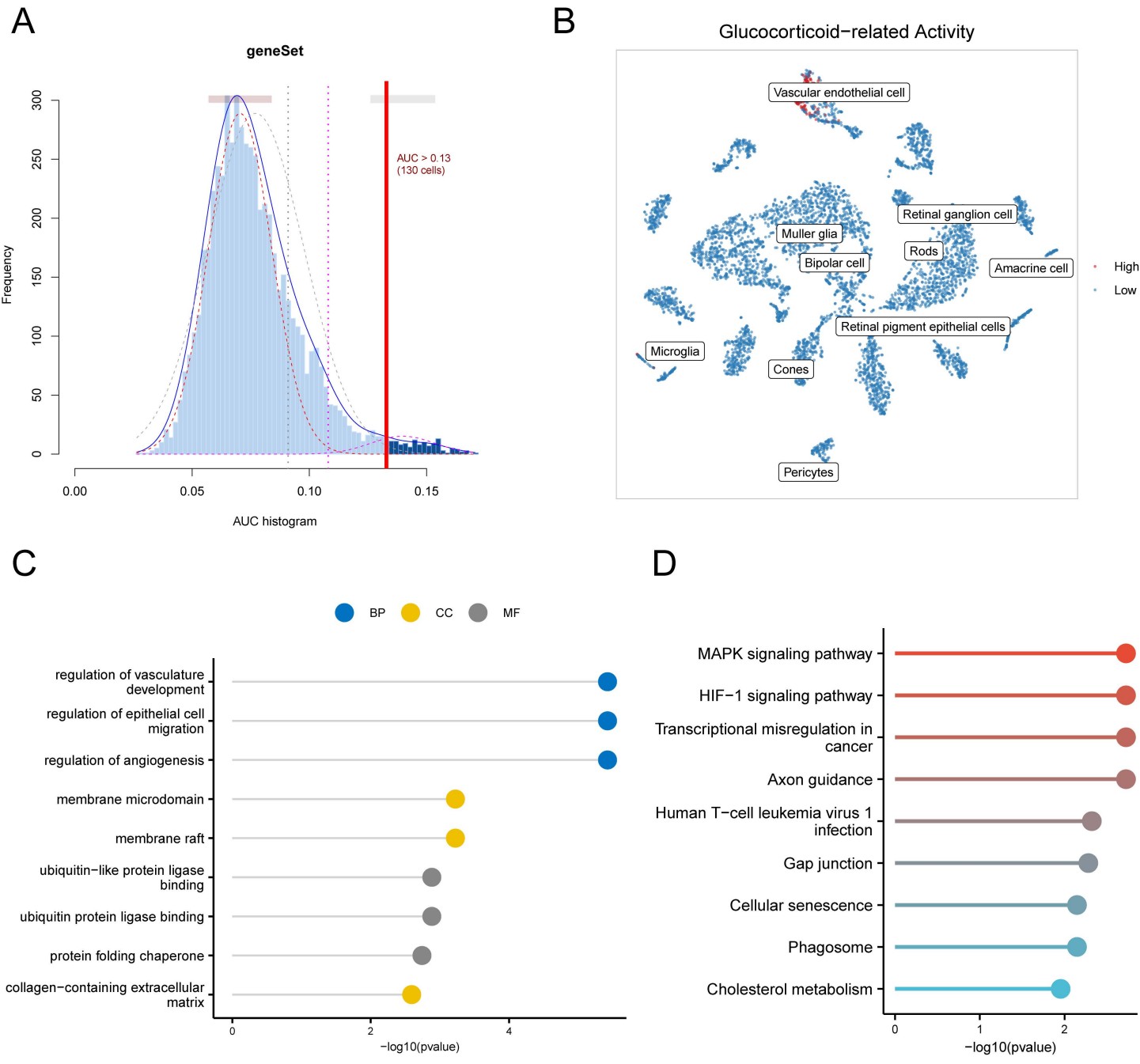

**Fig 3. Identification of subgroups of glucocorticoid activity in DR.** (A) Frequency of cell distribution at different AUC values. (B) t-SNE plot demonstrating the localization of glucocorticoid-active cells. (C) GO-enriched lollipop plot. (D) KEGG-enriched lollipop plot.

## Cell-cell communication analysis

In order to uncover the crosstalk between the various cell types in the DR, we examined more thoroughly their interaction networks. Our results show that even as the number of interactions rose, the intensity of interactions between cell types

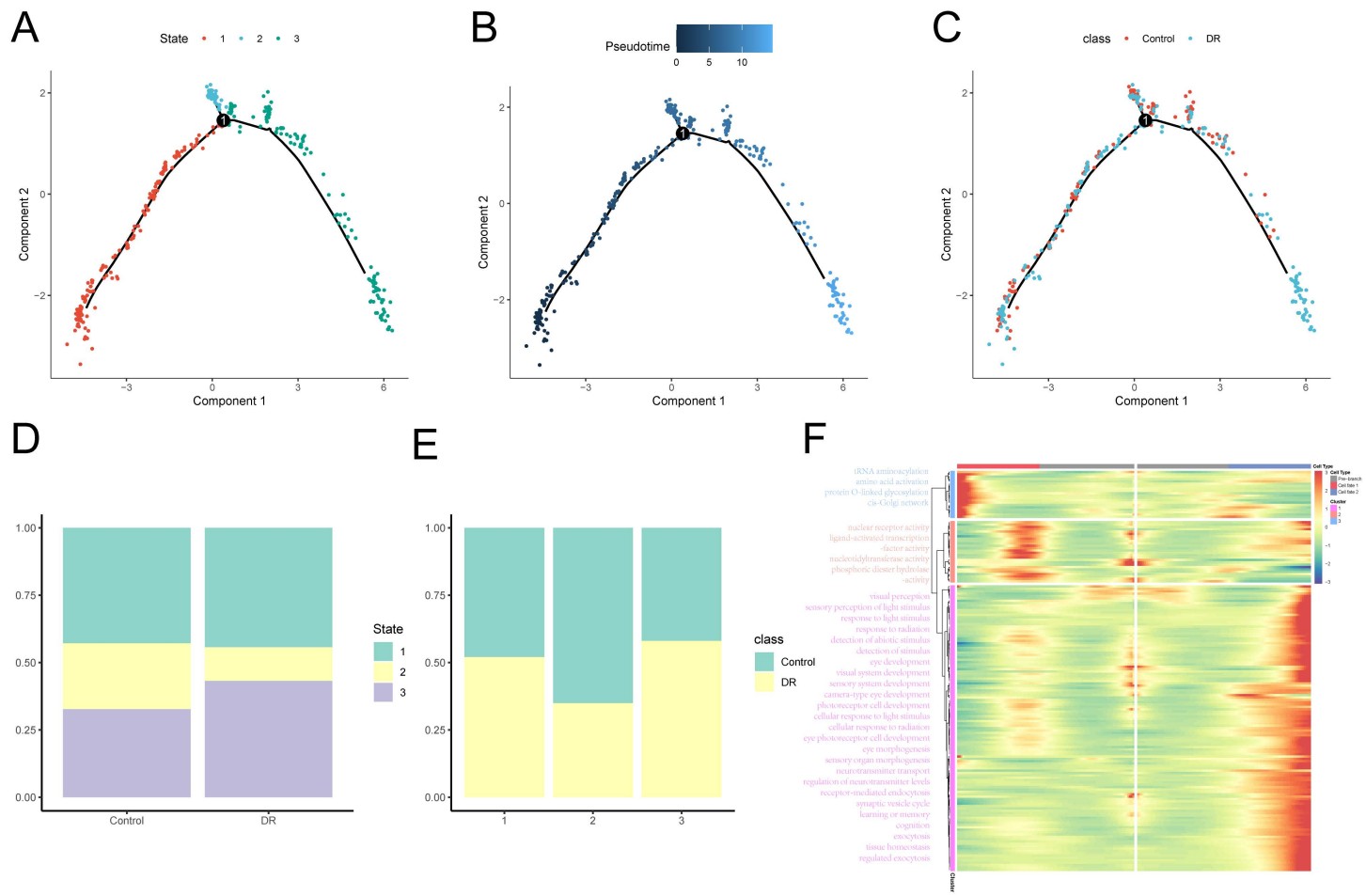

**Fig 4. Pseudotime analysis reveals the differentiation pattern of VECs in DR and suggests potential functional differences.** (A) Monocle2 categorizes the pseudotime trajectory of VECs into 3 different states. (B) The direction of differentiation of the pseudotime trajectory is shown with a color gradient transitioning from dark blue to light blue, indicating a deferral of differentiation from the starting point to the end point. (C) Distribution of VECs in DR and Control groups on the pseudotime trajectory. (D) Stacked histogram demonstrating the distribution of different States in the DR group versus the Control group. (E) Stacked histograms demonstrating the distribution of VECs in different States in DR group vs Control group. (F) The heatmap illustrates DEGs across various branches (cell fates). GO pathways that are significantly enriched in the distinct gene clusters depicted in the heatmap are presented on the left.

in the DR group decreased in comparison to the Control group (Fig 6A). Focusing on VECs as the initiators of communication, we visualized intergroup communication in Fig 6B, C, revealing that while the majority of interaction exhibited no discernible change, communication between VECs and amacrine cells (ACs) exhibited a marked increase in both number and strength in the DR group compared to the Control group. The overall signaling patterns between the Control and DR groups were then further compared, and Fig 6D clearly illustrates these patterns. It can be seen that the ANGPTL signaling intensity of VECs was significantly altered in the DR group, implying that this pathway may be related to its altered glucocorticoid activity.

Next, because of the significant relationship between VECs and glucocorticoid-related activity, we focused on exploring communication signals that were significantly altered between groups with the VECs as the initiator of communication. We analyzed receptor-ligand pairs that may regulate communication between VECs and other cells. PTN and ANGPTL signaling pathways are important signaling pathways DR. PTN and ANGPTL ligands derived from VECs bind to the

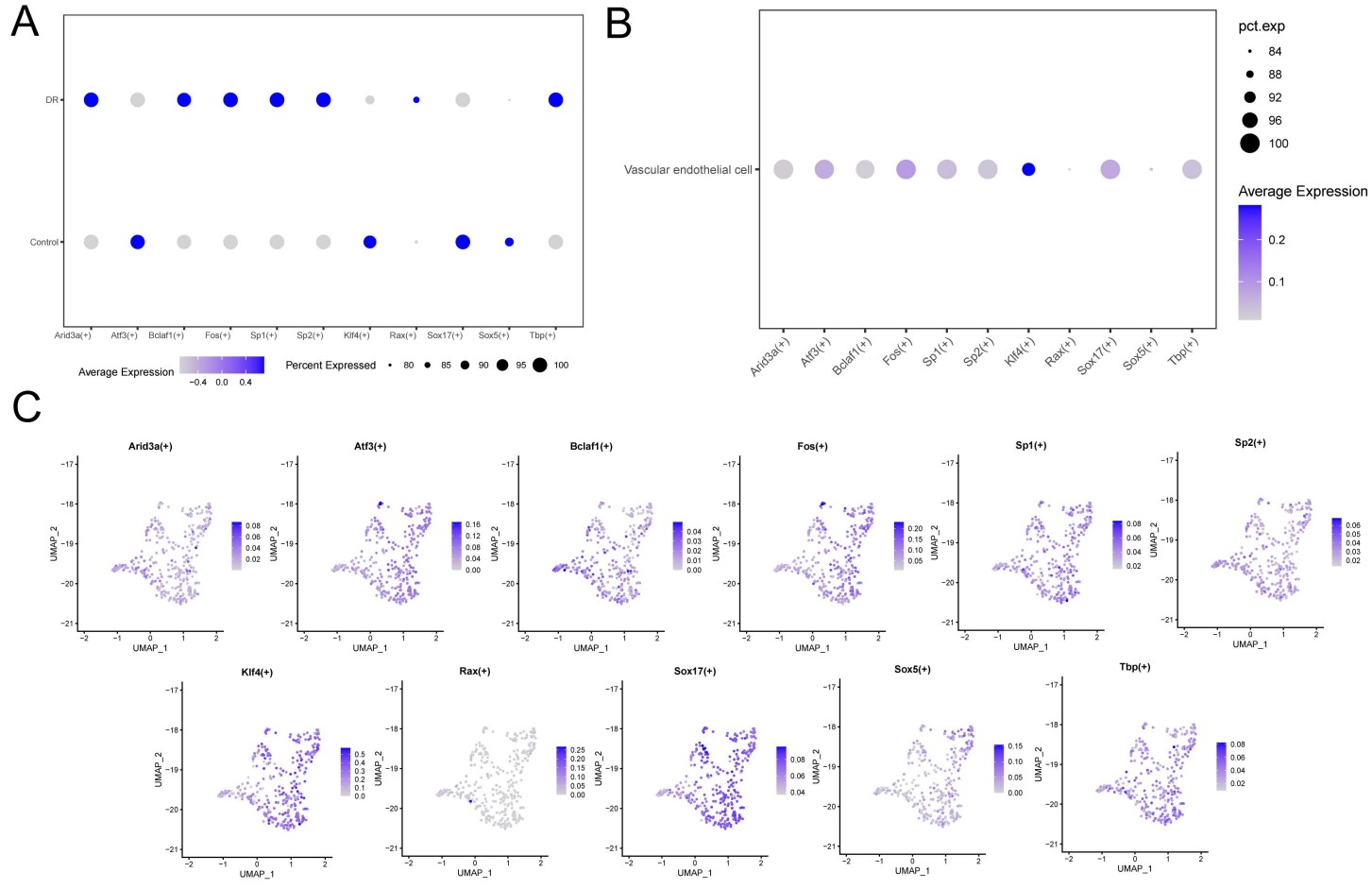

**Fig 5. Analysis of TFs in VECs.** (A) Bubble plots of 11 TFs in VECs that changed significantly between the two groups. (B) Bubble diagram of 11 TFs that changed significantly in all VECs. (C) Distribution of TF expression in all VECs.

**Table 1. Small Molecule Drug Screening.**

| TF | Drug | Interaction score |
|---|---|---|
| Atf3 | MECAMYLAMINE | 0.6251 |
| Fos | MAGNESIUM SULFATE ANHYDROUS | 1.0502 |
| Klf4 | HYDROXYUREA | 2.5004 |
| Tbp | ETOPOSIDE | 0.5933 |

corresponding receptors on ACs and muller glia cells, respectively, and this interplay is increased in DR. PTN ligands derived from VECs bind to the corresponding receptors on rods, muller glia cells, and microglia, and this interplay is reduced in DR (Fig 6E).

Additionally, We judged the expression degrees of genes associated with the PTN and ANGPTL pathways in several cell types. Interestingly, VECs in the DR group had much more PTN ligand expression than the Control group, and retinal ganglion cells, bipolar cells, and ACs all had increased receptor NCL expression. This finding elucidates the establishment of the PTN signaling pathway between VECs and retinal ganglion cells, bipolar cells, and ACs in tissues affected by DR.

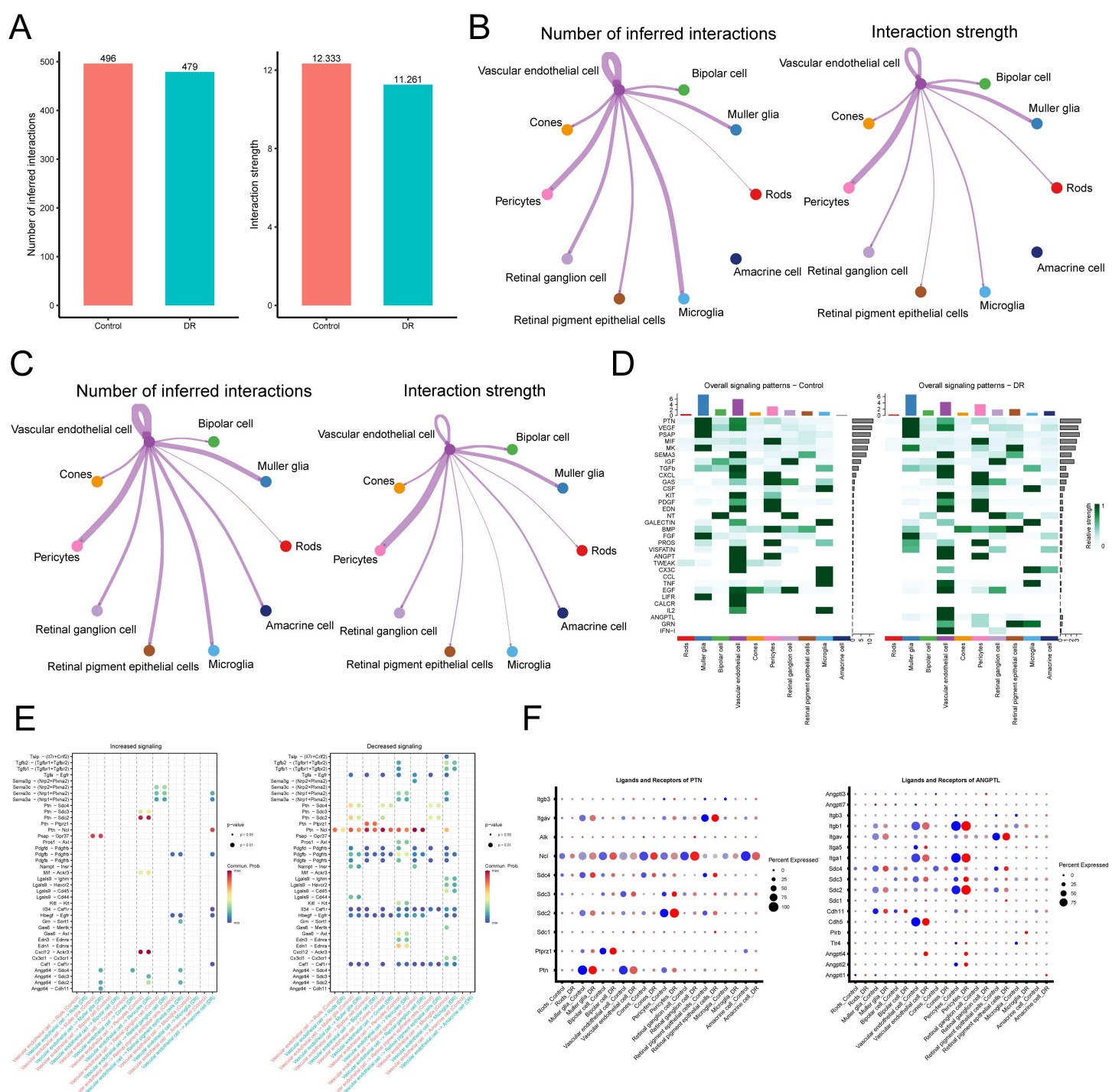

**Fig 6. Cell-cell communication in Control and DR groups.** (A) Bar graph illustrating the cumulative quantity and overall intensity of interactions among all cell types within individual cells in both the Control group and the DR group. (B) Network graphs demonstrating the number and intensity of communication signals of VECs as communicating participants in Control group, with all other cells. (C) Network graphs demonstrating the number and intensity of communication signals of VECs as communicating participants in DR group, with all other cells. (D) Signaling pathways with the largest overall signaling pathway contribution in the Control and DR groups. (E) Ligand-receptors with increased and decreased communication strength between VECs and other cellular subpopulations in DR. (F) Expression distribution of ligands and receptors within PTN and ANGPTL signaling in Control and DR groups. Light blue to dark red is the color gradient's transition, signifying modest to high expression.

Similarly, a much higher expression of ANGPTL ligands was tested in VECs from the DR group when juxtaposed with the Control group, whereas receptors such as Sdc2, Sdc4, and Cdh11 demonstrated elevated expression levels in muller glia cells, among others. The reason for the formation of ANGPTL pathway formation between VECs and muller glia cells in DR tissues was illustrated (Fig 6F).

## Hub gene screening

In our hub gene screening process, we focused on the 119 DEGs of VECs across the groups, which were submitted to the String database for PPI analysis, with results depicted in Fig 7A. Twenty-five hub genes were obtained from the 119 DEGs based on 12 CytoHubba algorithms (Fig 7B, S7 Table).

We performed GO and KEGG enrichment analysis to elucidate the biological roles of these hub genes. The GO results (Fig 7C, S8 Table) showed that these genes have a significant role in regulation of blood vessel endothelial cell migration, regulation of endothelial cell migration, endothelial cell migration (biological process, BP); membrane microdomain, membrane raft, early endosome (cellular component, CC); heat shock protein binding, protein folding chaperone, cell adhesion molecule binding (molecular function, MF) were enriched. These genes were significantly enriched in particular pathways, such as the Hypoxia-inducible factor-1 (HIF-1) signaling pathway and the estrogen signaling system, according to the findings of the KEGG pathway analysis (Fig 7D, S9 Table).

We also analyzed the correlations among the 25 hub genes (Fig 7E), and the heatmap results showed significant positive correlations among most of the hub genes, with the *ApoE* gene being significantly negatively correlated with most of the other hub genes.

## Hub gene regulatory network construction

First, we converted 25 mouse-derived hub genes to human-derived hub gene format. We used the StarBase online database to look for and register the matching mRNA/RBP pairings for 23 of the hub genes due to RNA binding proteins (RBPs) interact with mRNA. We created an RBP-mRNA network using the associations from the online dataset, which comprised 94 nodes, including 71 RBPs, 23 mRNAs, and 408 edges (Fig 8A). Next, We built a network of mRNA-miRNA-lncRNA interactions to clarify the possible molecular pathways involving hub genes. The data identified 12 hub genes as target mRNAs. the constructed mRNA-miRNA-lncRNA network included 96 nodes, 3 lncRNAs, 12 mRNAs, 81 miRNAs, and 245 edges (Fig 8B). Finally, we used the hTFtarget mouse database to find TFs that bind to important genes. Using Cytoscape software, we were able to display the data we collected on the interactions between 50 TFs and 16 hub genes (Fig 8C).

## Validation of Hub genes

We chose to treat HRMECs with different concentrations of glucose to verify the expression differences of DUSP6, AP1S2, and PTPRB. qRT-PCR results showed that after high-glucose treatment, the expression of AP1S2 was significantly upregulated, while the expressions of DUSP6 and PTPRB were significantly downregulated ($P < 0.05$) (Fig 9A–C).

## Discussion

DR is a prevalent complication among individuals with diabetes, characterized by pathological alterations in the retinal microvasculature. Significant intercellular contact between retinal pigment epithelial cells and other retinal cells was found in a prior investigation [27]. Our study found that glucocorticoid activity was significantly enhanced in DR tissues, especially in retinal VECs. Retinal VECs are pivotal in maintaining retinal vascular functionality and are integral to the pathogenesis of DR, aligning with findings from previous studies [28].

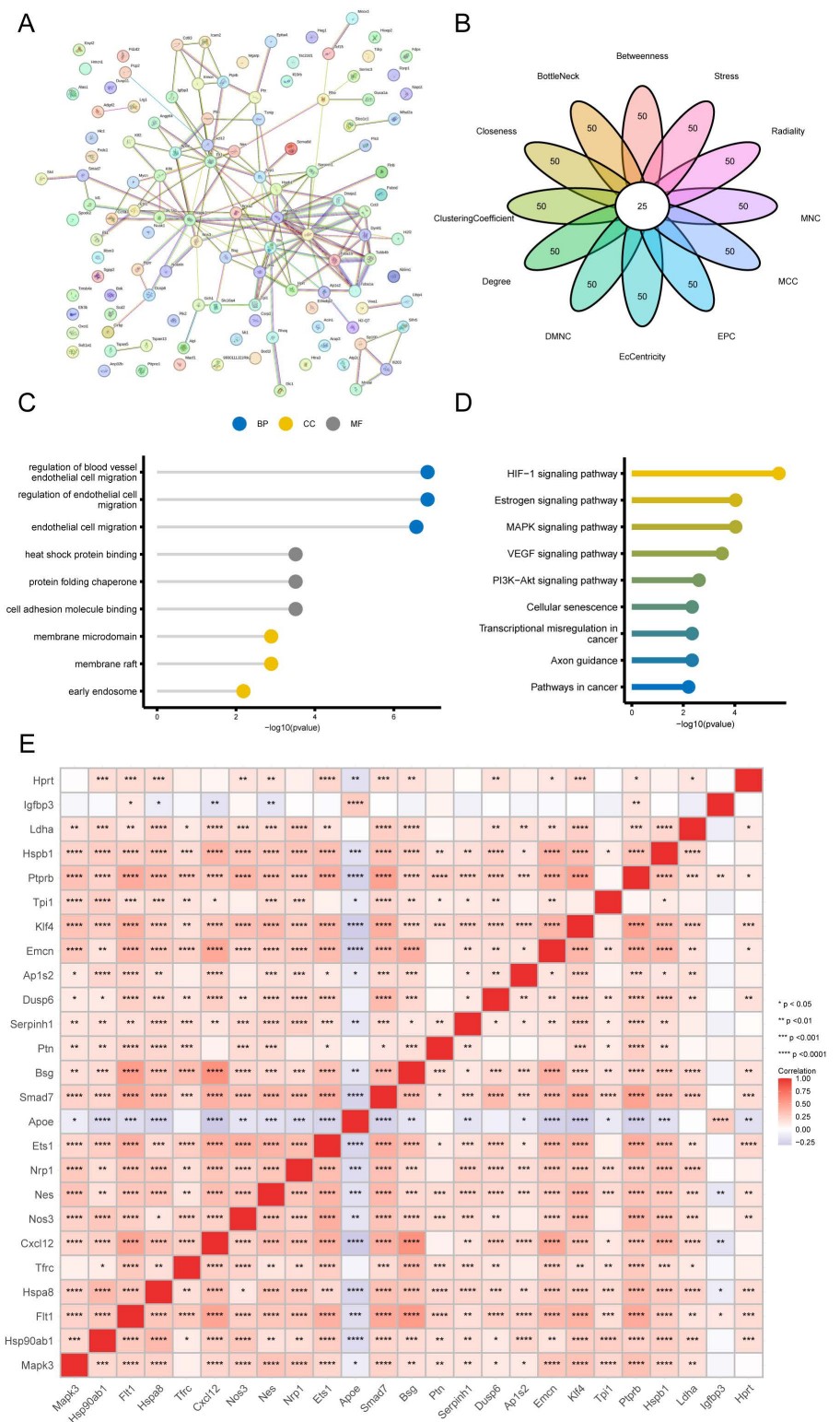

**Fig 7. PPI analysis, enrichment analysis and correlation analysis of DEGs.** (A) PPI analysis of DEGs. (B) 12 machine learning algorithms to screen hub genes. (C) GO enrichment. (D) KEGG enrichment. (E) Correlation analysis between hub genes.

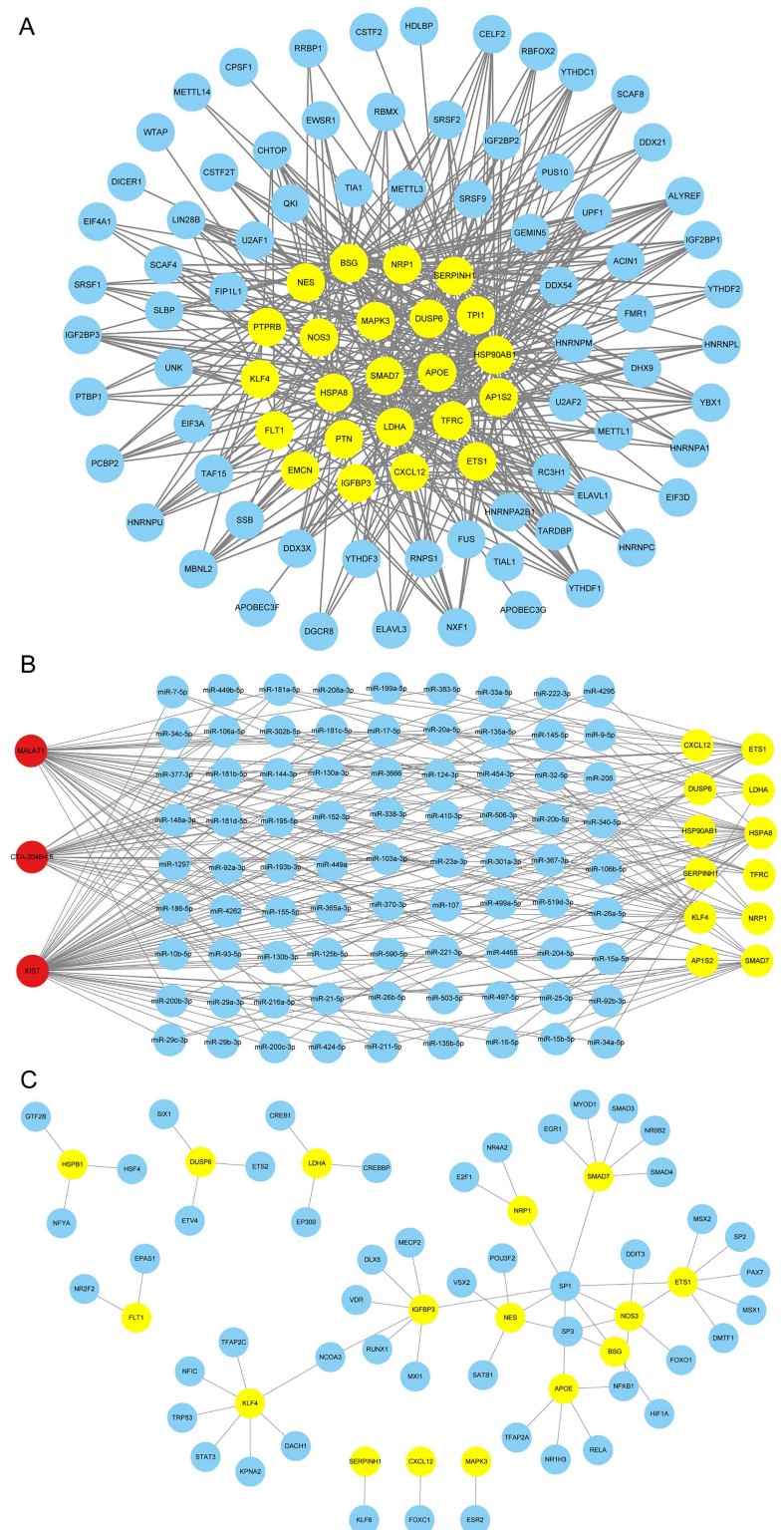

**Fig 8. Hub gene regulatory network.** (A) Hub gene regulation network of RBP-mRNA. Blue spheres represent RBP and yellow spheres indicate mRNA. (B) mRNA-miRNA-lncRNA network of hub genes. Blue spheres signify miRNAs, yellow spheres denote mRNA, and red spheres indicate lncRNA. (C) mRNA-TF interaction network of hub genes. Yellow spheres signify mRNA and blue spheres represent TF.

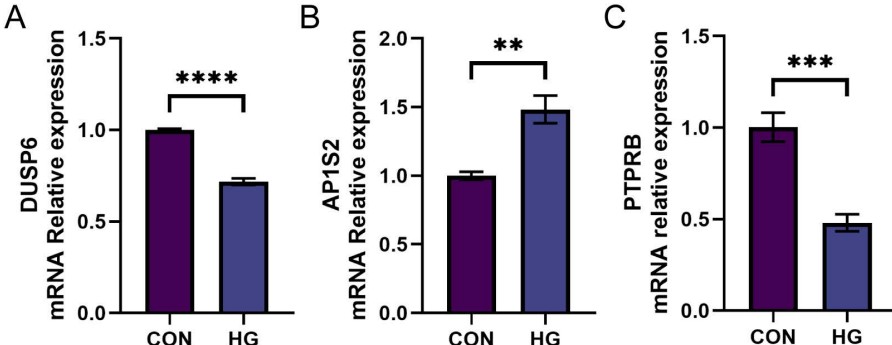

**Fig 9. Validation of Partial Hub Genes.** (A) qPCR was used to detect the relative expression level of DUSP6 mRNA in the two groups. (B) qPCR was used to detect the relative expression level of AP1S2 mRNA in the two groups. (C) qPCR was used to detect the relative expression level of PTPRB mRNA in the two groups. *P<0.05; **P<0.01; ***P<0.001; ****P<0.0001.

In the study of DR, we revealed the transcriptional status of retinal VECs in different branching states and their biological functions by mimetic temporal analysis. According to our results, it is possible that retinal VECs' unique expression patterns in cell differentiation and functional regulation are closely related to how DR develops. Our study found that in DR, retinal VECs exhibit two major branches. genes highly expressed in State 1 are involved in nuclear receptor activity, which may be related to cellular response to glucocorticoids. Nuclear receptors such as glucocorticoid receptors play important roles in cellular responses to metabolism and inflammation [29]. In DR, activation of these receptors may lead to functional changes in VECs, which in turn promote pathologic angiogenesis and retinal damage [30]. In addition, we observed upregulation of genes associated with amino acid activation and protein O-linked glycosylation in State 2, which may be related to the metabolic state and viability of the cells. Metabolic reprogramming of endothelial cells under DR conditions has been suggested to be an important mechanism for promoting cell survival and resistance to apoptosis [31]. This metabolic adaptation may be an important aspect of DR development and, by maintaining endothelial cell survival, may contribute to slowing disease progression. Notably, in State 3, the enriched genes were associated with visual perception and light stimulus responses. Curiously, VECs are not directly involved in visual perception, which primarily involves photoreceptor cells and visual pathways in the retina. However, VECs contribute to the preservation of the retinal vasculature, ensuring that the retina receives an adequate blood supply, which is possibly necessary for visual perception. Research indicates that VECs can respond to photostimulation through red or near-infrared light exposure, facilitating angiogenesis via a series of biochemical mechanisms, including extracellular Ca2+ influx, progression through the cell cycle, cell proliferation, nitric oxide (NO) production, and VEGF synthesis [32]. It has also been shown that light-responsive pathways are essential for promoting retinal vascularization in late-gestation mice during eye development [33]. In conclusion, while VECs do not directly engage in visual or sensory perception, they are vital for preserving the health of retinal blood vessels, which might be critical for visual acuity based on bioinformatics analysis results. Moreover, they exhibit certain responses to light stimuli that may be linked to processes of angiogenesis and tissue repair.

Kruppel-like factor 4 (KLF4) is a zinc-finger TF that is indispensable for controlling cell division and death [34]. It has been demonstrated that KLF4 protects blood vessels. For example, it attenuates cerebrovascular injury by decreasing inflammation in VECs and controlling the production of tight junction proteins [35]. Additionally, KLF4 has been thoroughly investigated in relation to diabetes and its consequences. For instance, in human umbilical VECs exposed to high glucose, overexpression of KLF4 has been demonstrated to suppress the expression of proteins involved in autophagy and apoptosis [36]. It has also been shown that KLF4 improves foot cell function and reduces proteinuria by selectively regulating the epigenetic profile of foot cells, suppressing mesenchymal gene expression and enhancing epithelial gene

expression [37]. In particular, specific *KLF4* knockdown in the retina was found to result in elevated pericyte bridge density [38]. These results indicate that KLF4 might have an important position in DR, and elucidating its precise mechanisms could yield valuable insights into potential therapeutic interventions for this condition. Hydroxyurea is an antineoplastic agent commonly employed in the management of sickle cell anemia, myeloproliferative disorders, and various malignancies [39]. In our study, it was predicted to bind to KLF4. Studies have shown that there is a positive effect of hydroxyurea on VECs in the appropriate dose range [40]. Treatment with hydroxyurea has been demonstrated to lower intravascular hemolysis ranks and markers of impaired endothelial function in sickle cell anemia patients [41]. Moreover, hydroxyurea has been shown to protect diabetic cardiomyopathy by preventing inflammatory processes and apoptosis, and it demonstrates anti-angiogenic qualities in tumor tissues both in vitro and in vivo [42]. Collectively, the vasoprotective and anti-angiogenic attributes of hydroxyurea underscore its potential as a therapeutic candidate for diabetic retinopathy.

In our study of DR using scRNA-seq data, we identified significant alterations in metabolic pathways that have important implications for understanding the molecular mechanisms underlying DR. We specifically examined the roles of pleiotrophin (PTN) and angiopoietin-like proteins (ANGPTLs) in the pathology of DR. PTN, a multifunctional cytokine, has been implicated in various angiogenic and fibrotic diseases. PDR patients were shown to have greater vitreous PTN the quantities than non-diabetic controls. The expression of PTN was particularly heightened.in the fibrovascular membranes of affected individuals. According to functional studies, human umbilical VECs and retinal pigment epithelial cells experienced increased apoptosis and decreased cell proliferation, migration, and angiogenesis when PTN was silenced in hyperglycemic and hypoxic environments. Mechanistically, VEGF was downregulated and ERK 1/2 phosphorylation decreased when PTN was reduced. This suggests that PTN may stimulate angiogenesis and cell proliferation via controlling the VEGF and ERK 1/2 signaling pathways [43]. Interestingly, we discovered that, as compared to the control group, the DR group had a higher number and strength of communication signals between VECs and ACs, as well as a substantial increase in PTN-NCL signals between VECs and ACs. ACs are key interneurons in the inner layer of the retina. The name derives from the Greek "a-" (without) and "makros" (long), referring to their lack of typical axons [44]. According to clinical research, diabetic patients' retinal electroretinogram (ERG) shows early oscillatory potentials (OPs) alterations before the fundus examination can detect DR [45]. OPs are considered to be the main indicator of ACs function, suggesting that ACs damage may be the earliest event in DR. In many illness models, PTN has been demonstrated to protect neurons [46]. Following its release by VECs, PTN attaches itself to NCL on the membrane surface of ACs, perhaps halting their apoptotic process. Park et al. discovered that anti-VEGF therapy of the vitreous cavity in diabetic rats accelerated the apoptosis of ACs in the inner nuclear layer [47]. As a result, PTN-NCL signaling may enhance ACs survival by modulating VEGF. Similarly, ANGPTL-4, another key player in DR, has been shown to modulate inflammation and angiogenesis. Both diabetic retinas and human retinal VECs addressed to high glucose have noticeably higher levels of ANGPTL-4. HIF-1α activation was necessary for this upregulation to occur. ANGPTL-4 has been shown to enhance inflammation, increase permeability, and promote angiogenesis through the profilin-1 signaling pathway. These findings highlight ANGPTL-4 as a key regulator of diabetic microvascular dysfunction and retinal inflammation, indicating that DR may benefit from treatment benefits that target this pathway [48]. Furthermore, the vitreous fluid of PDR patients showed the interaction of several angiogenic and inflammatory mediators, such as IL-8, VEGF, and ANGPTL-4. The strong correlations between these mediators suggest that inflammatory and angiogenic interact in the pathophysiology of PDR. Moreover, the concentrations of these mediators were associated with renal impairment, indicating shared pathways in DR and nephropathy [49]. In summary, our study underscores the important roles of PTN and ANGPTL-4 in the advancement of DR. The identification of these pathways not only enhances our comprehension of the molecular mechanisms underlying DR but also points to prospective therapeutic intervention aims. Future studies in the setting of DR should focus on verifying these results in bigger cohorts and examining the possible therapeutic benefits of targeting the PTN and ANGPTL-4 pathways.

Our study identified 25 hub genes that play significant roles in DR, closely linked to the regulatory network of glucocorticoid signaling. The association of these genes with glucocorticoids is manifested in multiple physiological processes, including cell signaling, stress response, inflammation regulation, and metabolic regulation. Glucocorticoids inhibit neovascularization by affecting *FLT1* and *NRP1* in the VEGF signaling pathway [50,51]. Inhibition of *CXCL12* reduces inflammatory cell recruitment and endothelial cell migration [52]. In addition, heat shock proteins (*HSP90AB1*, *HSPA8*, *HSPB1*) are crucial in safeguarding cells against oxidative stress, inflammation, and apoptosis, while also maintaining the stability of the glucocorticoid receptor [53]. Diabetes problems have been linked to changes in the expression of the heat shock protein, which is functionally linked to cellular damage brought on by hyperglycemia [54]. Consequently, in order to lessen the negative consequences of hyperglycemia on the target organs implicated in diabetic vascular problems, heat shock proteins are crucial. Another critical gene, *NOS3*, is implicated in vasodilation and the preservation of the vascular network through the regulation of NO release from endothelial cells [55]. Glucocorticoids may avert microcirculatory disturbances in DR by upregulating NOS3 expression, thereby enhancing NO production [56]. ApoE, or Apolipoprotein E, is a gene integral to lipid metabolism and has been extensively studied concerning diabetic complications. Each of the three primary alleles—ε2, ε3, and ε4—encodes ApoE2, ApoE3, and ApoE4, respectively, and has unique roles and correlations with various diseases. Plasma ApoE protein levels are a risk factor for the development and severity of DR, according to the majority of clinical research [57,58]. Moreover, genetic predisposition to DR is unaffected by polymorphisms in the ApoE gene [59,60]. Nonetheless, some research indicates that particular ApoE alleles might affect how severe DR is. For instance, in the Czech population, carriers of the ε4 allele have been associated with a decreased risk of retinopathy in Type 2 diabetics [61]. Carriers of the ε2 allele are linked to a higher risk of DR among Brazilians [62]. DR individuals with the ε4 allele showed a strong trend toward visual impairment and more severe retinal hard exudates, according to another study on type 2 diabetic patients in Mexico [63]. The ApoE4 subtype has no influence on retinal neovascularization, although the ApoE2 and ApoE3 subtypes do, according to cell and animal studies [64]. According to this study, variations in ApoE subtypes may affect retinal neovascularization by controlling the function of endothelial cells. The ApoE4 subtype may have an inhibitory effect in contrast to ApoE2 and ApoE3, which is in line with the findings of clinical studies. Variations in population genetic backgrounds and observational measures may be the cause of differences in study outcomes, requiring more research on the functional variability of ApoE isoforms.

In this study, we identified three new hub genes associated with DR: dual specificity phosphatase 6 (*DUSP6*), adaptor protein 1 sigma 2 (*AP1S2*), and protein tyrosine phosphatase receptor type B (*PTPRB*). These findings, which have never been documented before, highlight how important these genes may be in the pathophysiology of DR. DUSP6, a dual-specificity phosphatase, plays a crucial role in regulating the MAPK signaling pathway [65]. Numerous clinical illnesses, including as cancer and cardiovascular ailments, have been linked to its dysregulation [66]. Research has indicated that DUSP6 can influence corneal epithelial proliferation by controlling ERK1/2 phosphorylation [67]. We hypothesize that the ERK pathway may be continuously activated in the context of DR due to the decreased expression of DUSP6, which would hasten the development of DR. AP1S2, a vital element of the adaptor protein complex 1, plays an indispensable role in clathrin-mediated endocytosis, a process crucial for the preservation of cellular equilibrium and signaling [68]. Pettigrew syndrome, which presents as X-linked intellectual impairment along with congenital hydrocephalus and basal ganglia calcification, can be brought on by an AP1S2 mutation [69]. AP1S2 might play a role in the disruption of the membrane protein endocytosis system in DR, subsequently influencing the permeability and inflammatory responses of retinal endothelial cells. Finally, PTPRB is a protein tyrosine phosphatase of receptor type that inhibits the VEGF signaling pathway, hence negatively regulating angiogenesis [70]. According to a study, in glaucoma models, blocking PTPRB can stop retinal ganglion cell layer destruction. This resembles the initial stages of DR, which primarily presents as damage to the ganglion cells [71]. This could mean that PTPRB is a neuroprotective factor in DR as well as an endothelial protective factor, and that declining levels could worsen the course of DR. In addition to expanding our knowledge of the

molecular landscape of DR, the identification of these hub genes creates new opportunities for tailored therapeutics that target these particular pathways, which may improve the prognosis of people with this crippling illness.

Our research identified a number of important signaling pathways associated with DR, most notably the HIF-1 signaling pathway, which is essential to the pathophysiology of the illness. The HIF-1 pathway is recognized for its regulation of genes that contribute to angiogenesis, metabolism, and cellular survival amid hypoxic environments, conditions frequently encountered in DR. An elevated amount of HIF-1α have been documented in the retinal tissues of diabetic individuals, resulting in an overproduction of VEGF, a factor that stimulates detrimental neovascularization and enhances vascular permeability [72]. In our study, we observed significant enrichment of the HIF-1 signaling pathway, indicating its crucial role in mediating hypoxia-related effects in DR. This is in line with earlier research showing how HIF-1α contributes to retinal inflammation and neovascularization, two important features of DR [73]. Estrogen is a vital hormone in female development and can influence cellular growth and metabolism through the regulation of gene expression [74]. Specifically, our research revealed a close association between the estrogen signaling pathway and the effects of glucocorticoids in DR. The first possible reason is that both are steroid hormones secreted by the adrenal cortex, and their receptors are often activated at the same time, possibly in crosstalk. The second reason is that several illnesses, including diabetes, have been connected to the onset and progression of estrogen activity disturbances or dysregulation [75]. Estrogen has demonstrated protective effects against apoptosis in pancreatic β-cells. Male mice have lower insulin concentrations and fewer pancreatic β-cells than female mice, and their cells are more vulnerable to oxidative stress. It was also found that estrogen treatment also enhanced insulin secretion in the pancreatic islets of both diabetic and non-diabetic participants [76]. These results suggest that drugs that activate the estrogen pathway have therapeutic potential for DR.

## Conclusion

To sum up, this study skillfully used scRNA-seq data to examine how glucocorticoid-related genes function in DR. Significant outcomes include the identification of DEGs, critical metabolic pathways, and essential TFs, alongside the establishment of various regulatory networks. These findings highlight possible treatment targets and provide a foundation for comprehending the molecular mechanisms underlying DR. Moving forward, integrating these insights with clinical data and conducting experimental validation could facilitate the development of innovative treatment strategies for DR, ultimately enhancing patient outcomes.

## Supporting information

**S1 Table.  Glucocorticoid-related gene.**
(TXT)

**S2 Table.  Cell biomarkers.**
(TXT)

**S3 Table.  DEGs.**
(CSV)

**S4 Table.  GO enrichment analysis of DEGs.**
(CSV)

**S5 Table.  KEGG enrichment analysis of DEGs.**
(CSV)

**S6 Table.  GO enrichment analysis of various states.**
(CSV)

**S7 Table. Hub genes.**
(TXT)

**S8 Table. GO enrichment analysis of Hub genes.**
(CSV)

**S9 Table. KEGG enrichment analysis of Hub genes.**
(CSV)

## Author contributions

**Data curation:** Rong Zhang, Lin Wang, Yongrui Wang, Xiaodan Zhang.

**Supervision:** Guohong Zhou.

**Writing – original draft:** Lingda Wang.

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
