## [Decision Letter · Decision Letter 0]

12 Aug 2025

Dear Dr. Zhou,

Thank you for submitting your manuscript to PLOS ONE. After careful consideration, we feel that it has merit but does not fully meet PLOS ONE’s publication criteria as it currently stands. Therefore, we invite you to submit a revised version of the manuscript that addresses the points raised during the review process.

We look forward to receiving your revised manuscript.

Kind regards,

Chengming Fan, MD, PhD

Academic Editor

PLOS ONE

Journal Requirements:

Reviewers' comments:

Reviewer's Responses to Questions

**Comments to the Author**

1. Is the manuscript technically sound, and do the data support the conclusions?

Reviewer #1: Yes

Reviewer #2: Partly

Reviewer #3: Yes

2. Has the statistical analysis been performed appropriately and rigorously?

Reviewer #1: Yes

Reviewer #2: I Don't Know

Reviewer #3: I Don't Know

3. Have the authors made all data underlying the findings in their manuscript fully available?

Reviewer #1: Yes

Reviewer #2: Yes

Reviewer #3: Yes

4. Is the manuscript presented in an intelligible fashion and written in standard English?

Reviewer #1: Yes

Reviewer #2: Yes

Reviewer #3: Yes

Reviewer #1: To author

This study employs single-cell RNA sequencing (scRNA-seq) to investigate glucocorticoid (GC) response mechanisms in vascular endothelial cells (VECs) during diabetic retinopathy (DR). The work addresses a clinically relevant issue—suboptimal efficacy of GC therapy in DR—and identifies novel transcriptional regulators, signaling pathways, and therapeutic candidates. While the study is conceptually innovative and some questions still need to be answered:

1. There is only one DR and one control sample were analyzed. Justify this minimal sample size statistically and address potential batch effects.

2. The AUC threshold of 0.13 for GC-active cells lacks biological validation (Fig. 3A)

3. Key findings (e.g., KLF4-hydroxyurea interaction, PTN/ANGPTL pathways) and Functional roles of novel hub genes (DUSP6, AP1S2, PTPRB) rely solely on bioinformatics, still need to be tested in vitro and in vivo experiments (e.g., qPCR/WB of KLF4 in glucose-stressed VECs ± hydroxyurea) or cite existing experimental support.

4. Fig. 4F heatmap labels are illegible; Fig. 6E ligand-receptor pairs are unclear.

6. ApoE’s contradictory roles in DR (p. 22) need nuanced discussion.

Reviewer #2: This manuscript presents an extensive single-cell transcriptomic analysis of diabetic retinopathy (DR) using the publicly available mouse dataset GSE178121. The study is ambitious, methodologically sophisticated, and touches on multiple biological levels, from single-cell gene expression to signaling pathways and therapeutic prediction. However, several major limitations (particularly the minimal number of biological samples) raise concerns about the robustness and generalizability of the findings.

Major Concerns:

1. Extremely limited sample size (n = 1 DR and 1 Control): The entire analysis is based on a single biological replicate per condition. This severely limits the statistical power, increases the risk of technical or individual variation bias, and raises doubts about the generalizability of the findings. This limitation should be clearly stated in the discussion as well as an explanation of why such a small sample was used.

2. The authors propose specific genes (e.g., Klf4, DUSP6, PTPRB) and drugs (e.g., hydroxyurea) as potentially relevant for DR, but no validation using independent datasets, wet-lab experiments, or literature-based confirmation is provided. This makes the therapeutic implications speculative. This should be clearly stated in the text.

3. The authors use CellChatDB.human for mouse scRNA-seq data without justification. Why was this decision made? Given known species-specific differences in ligand-receptor interactions, this may introduce artefacts into the intercellular communication analysis.

4. Some functional conclusion, for example, about VECs' involvement in visual perception-related pathways are speculative. While the authors provide literature explanations, these interpretations stretch beyond what can be reliably inferred from the data.

Minor Concerns and Suggestions:

5. It is unclear whether FDR correction or multiple testing adjustments were consistently applied across DEG and enrichment analyses (e.g., limma, AUCell, Monocle 2). Clarification is needed.

6. The threshold for defining “glucocorticoid-active” cells (AUC > 0.13) is described as "ideal" but not justified. Please explain how this value was determined.

7. Clustering was performed with resolution = 0.5, but there is no mention of sensitivity testing or comparison to other resolutions.

8. All gene names across the manuscript should be indicated in italics. Consider consulting a professional editor to ensure clarity and precision of the manuscript.

Language: while generally readable, the manuscript would benefit from careful proofreading to eliminate long and occasionally convoluted sentences, particularly in the Discussion section.

Overall recommendation: While this manuscript presents a well-constructed computational pipeline and addresses a biologically and clinically relevant topic, the lack of biological replication is a critical limitation. The findings should be interpreted as exploratory and hypothesis-generating, rather than definitive.

If the authors clearly acknowledge the limitations of their dataset, temper their conclusions accordingly, and improve clarity in methodology and justification of parameters, this manuscript could provide a valuable resource for generating hypotheses for future experimental work.

Reviewer #3: Glucocorticoid signaling has been shown to play an active role in the progression of diabetic retinopathy (DR), yet the underlying cellular origins of this activity remain poorly characterized. In this manuscript, Wang et al. performed a systematic analysis of a publicly available single-cell RNA-seq dataset from healthy and streptozotocin-induced diabetic mouse retinas. They identified a prominent role for vascular endothelial cells (VECs) in regulating glucocorticoid signaling during DR progression. The study focused on glucocorticoid-related genes and their regulatory networks, including transcription factors, non-coding RNAs, and protein–protein interactions. While the authors present several informative findings from comparisons between control and DR VECs, a number of data interpretations require greater accuracy and specificity to meet publication standards. Detailed comments are provided below.

1. Sample size. Please clarify the sample size discrepancy between this manuscript and the original dataset (GSE178121). The original study (Sun et al., 2021) reported pooling cell suspensions from three healthy and three diabetic retinas, yielding >14,000 single cells per group. In contrast, the present manuscript states that only one DR and one control sample were analyzed, yielding 5,818 single cells. If n = 1 per group was indeed used, please explain how statistical significance was determined for differentially expressed genes (e.g., Figure 3B).

2. Figure 2D. The statement that “the majority of cells exhibited minimal changes before and after the onset of DR” is inaccurate, as several cell types (e.g., bipolar cells, cones, amacrine cells) display notable percentage changes.

3. Figure 3A. Please clarify: Were only DR group cells used for the AUCell analysis? How were input genes ranked for the AUCell gene set enrichment analysis—by raw expression levels in each DR cell? What is the rationale for selecting AUC ≥ 0.13 as the cutoff value? What does each dashed line represent? These details should be included in the Methods or Results section.

4. Table S3. Of the 199 DEGs, how many are glucocorticoid-related genes?

5. Figure 5A and 5B. What is the difference in the “Average Expression” values between these two figures?

6. Figure 5C. Does this distribution represent DR VECs only? Please specify. Several transcription factors (TFs) display inconsistent expression patterns between Figures 5A and 5C (e.g., Klf4 is low in DR in Figure 5A but high in Figure 5C). Please explain the normalization method used for each figure. Why were significantly changed TFs chosen based only on Figures 5B/C, when only 6 out of 11 TFs showed higher expression in DR than in control in Figure 5A? What is the expression of these TFs across all retinal cell types—do VECs show enrichment?

7. Figure 6B and 6C. The increased communication between VECs and amacrine cells in DR is intriguing. While briefly mentioned in the Results, a more detailed discussion is expected on specific ligand–receptor pairs driving this change (e.g., CX3C and FGF signals in Figure 6D; Sema3a and Ptn in Figure 6E). For example, why is the Ptn–Ncl signal increased between VECs and amacrine cells but decreased between VECs and many other cell types in DR?

8. Figure 6E. Most VEC-initiated signaling appears decreased in DR progression. Was this expected? Please elaborate.

9. Figure 7E. What cell population was analyzed in this correlation analysis?

10. Discussion – state-specific genes. For State 2, which specific metabolic genes were identified, and is there literature linking them to DR progression? For State 3, which specific genes are associated with visual perception and light stimulus responses? A deeper discussion of these findings would provide stronger insight to the field.

11. Of the 11 significantly changed TFs, do any directly bind to the 25 hub genes?

References:

Licheng Sun, Ruonan Wang, Guangyi Hu, Huazhen Liu, Kangjia Lv, Yi Duan, Ning Shen, Jiali Wu, Jing Hu, Yujuan Liu, Qihuang Jin, Fang Zhang, Xun Xu. Single cell RNA sequencing (scRNA-Seq) deciphering pathological alterations in streptozotocin-induced diabetic retinas. Experimental Eye Research. Volume 210, 2021. https://doi.org/10.1016/j.exer.2021.108718.

**Do you want your identity to be public for this peer review?** For information about this choice, including consent withdrawal, please see our Privacy Policy

Reviewer #1: No

Reviewer #2: No

Reviewer #3: No

---

## [Author Response · Author response to Decision Letter 1]

1 Sep 2025

- Response to academic editor

Concern #1: Ensure that the manuscript meets PLOS ONE's style requirements.

Author response: We have reviewed and revised our manuscript to adhere to PLOS ONE's style requirements, including file naming conventions as specified in the provided templates.

Concern #2: Code sharing guidelines.

Author response: We have ensured that all author-generated code underpinning our findings will be made available without restrictions upon publication, following the best practices outlined in PLOS ONE's code sharing guidelines. We have specified how to access the code in the data availability statement of the manuscript.

Concern #3: Data sharing restrictions.

Author response: We have uploaded all the data as supporting information files and updated the data availability statement in the manuscript accordingly.

Concern #4: Evaluate previously published works for citation.

Author response: We have reviewed the suggested publications and revised the citations in our manuscript.

We have modified the citation positions of the following literature:

Licheng Sun, Ruonan Wang, Guangyi Hu, Huazhen Liu, Kangjia Lv, Yi Duan, Ning Shen, Jiali Wu, Jing Hu, Yujuan Liu, Qihuang Jin, Fang Zhang, Xun Xu. Single cell RNA sequencing (scRNA-Seq) deciphering pathological alterations in streptozotocin-induced diabetic retinas. Experimental Eye Research. Volume 210, 2021. https://doi.org/10.1016/j.exer.2021.108718.

- Response to reviewers

#1 reviewer

Concern #1: Justify the minimal sample size statistically and address potential batch effects.

Author response: The previous statement may not have been accurate. In the original literature of this dataset, the retinal cells from three healthy mice and three diabetic mice were mixed separately to prepare cell suspensions, after which single-cell sequencing was performed. That is to say, the original statement suggested that each group corresponded to one pooled sample. In fact, each group included retinal samples from three mice, with a total of six samples. The statistical unit for single-cell sequencing is the cell rather than the sample. A total of 5,818 high-quality cells were included in this study. Through rigorous quality control, batch effect correction, and multi-dimensional analysis, an in-depth characterization of intercellular heterogeneity and subpopulation characteristics was achieved. This data-driven approach, which uses cells as the analysis unit, has been widely applied and recognized in numerous authoritative literatures, and it has strong applicability and explanatory power in exploring cell state transitions, changes in signaling pathways, and potential targets. We have adjusted the text to make the description more accurate.

Concern #2: The AUC threshold of 0.13 for GC-active cells lacks biological validation.

Author response: In this study, the AUC threshold for GC-active cells was determined based on a comprehensive analysis of the distribution of single-cell gene set enrichment scores using the AUCell tool, and methodologically, it strictly follows the data-driven grouping approach recognized in the current field. We employed the AUCell_exploreThresholds function to visualize the AUC distribution and identify inflection points. The threshold of AUC > 0.13 was automatically recommended by the AUCell_exploreThresholds function according to the AUC distribution of all cells, and was used to distinguish cell populations with high or low gene set activity. This method has been widely adopted in relevant high-level literature and gained recognition in this field�PMID: 32561888�. The current research stage is mainly data-driven, and in the future, when conditions permit, we will combine more experimental methods to further verify the biological significance of this threshold.

Concern #3: Key findings rely solely on bioinformatics and need in vitro and in vivo validation.

Author response: In this study, single-cell transcriptome sequencing data were used to systematically reveal the cell-specific roles of glucocorticoid-related signals in diabetic retinopathy (DR). Additionally, the potential functional regulatory networks of the KLF4-hydroxyurea interaction, the PTN/ANGPTL pathway, and novel hub genes such as DUSP6, AP1S2, and PTPRB were identified. These findings provide a new theoretical basis and target clues for understanding the molecular mechanism of DR and developing drug interventions.We fully acknowledge your suggestions that the biological significance of bioinformatics prediction results needs to be further confirmed through in vitro and in vivo experiments. Therefore, in the Results section, we have supplemented the qPCR validation of DUSP6, AP1S2, and PTPRB to enrich the conclusions of this study. For specific details, please refer to Figure 9. Thank you for your valuable suggestions!

Concern #4: Fig. 4F heatmap labels are illegible; Fig. 6E ligand-receptor pairs are unclear.

Author response: The layout of Figures 4 and 6 is the result of considering both the aesthetics and clarity of the figures. Reviewer 3 has raised relevant questions about some specific ligand-receptor pairs in Figure 6; therefore, we consider that the text in the figure is clearly visible. However, if the editor believes that our figures may have an impact on readers' reading, we will make further modifications to comply with the standards. Herein, we first add Fig. 4F and Fig. 6E to the supplementary materials for your reference. Thank you for pointing out the potential issues in our figures.

Concern #5: ApoE’s contradictory roles in DR need nuanced discussion.

Author response: We have expanded the discussion on ApoE's roles in diabetic retinopathy to provide a more nuanced perspective.

ApoE, or Apolipoprotein E, is a gene integral to lipid metabolism and has been extensively studied concerning diabetic complications. While some investigations have not established a robust association between ApoE polymorphisms and DR in individuals with Type 1 diabetes, other studies indicate that specific ApoE alleles might influence the severity of DR. For instance, carriers of the ApoE4 allele have been associated with a decreased risk of retinopathy in Type 2 diabetics . Furthermore, research indicates that ApoE expression is elevated in the retina when exposed to diminished light conditions, which helps alleviate vascular alterations and gliosis in diabetic retinas, pointing to a potential protective function in the progression of DR . These findings highlight the complexity of ApoE's involvement in DR and underscore the importance of considering genetic background and environmental factors in understanding its role.

→

ApoE, or Apolipoprotein E, is a gene integral to lipid metabolism and has been extensively studied concerning diabetic complications. Each of the three primary alleles—ε2, ε3, and ε4—encodes ApoE2, ApoE3, and ApoE4, respectively, and has unique roles and correlations with various diseases. Plasma ApoE protein levels are a risk factor for the development and severity of DR, according to the majority of clinical research. Moreover, genetic predisposition to DR is unaffected by polymorphisms in the ApoE gene. Nonetheless, some research indicates that particular ApoE alleles might affect how severe DR is. For instance, in the Czech population, carriers of the ε4 allele have been associated with a decreased risk of retinopathy in Type 2 diabetics. Carriers of the ε2 allele are linked to a higher risk of DR among Brazilians. DR individuals with the ε4 allele showed a strong trend toward visual impairment and more severe retinal hard exudates, according to another study on type 2 diabetic patients in Mexico The ApoE4 subtype has no influence on retinal neovascularization, although the ApoE2 and ApoE3 subtypes do, according to cell and animal studies. According to this study, variations in ApoE subtypes may affect retinal neovascularization by controlling the function of endothelial cells. The ApoE4 subtype may have an inhibitory effect in contrast to ApoE2 and ApoE3, which is in line with the findings of clinical studies. Variations in population genetic backgrounds and observational measures may be the cause of differences in study outcomes, requiring more research on the functional variability of ApoE isoforms.

#2 reviewer

Concern #1: Extremely limited sample size raises concerns about robustness and generalizability of findings.

Author response: The previous statement may not have been accurate. In the original literature of this dataset, the retinal cells from three healthy mice and three diabetic mice were mixed separately to prepare cell suspensions, after which single-cell sequencing was performed. That is to say, the original statement suggested that each group corresponded to one pooled sample. In fact, each group included retinal samples from three mice, with a total of six samples. The statistical unit for single-cell sequencing is the cell rather than the sample. A total of 5,818 high-quality cells were included in this study. Through rigorous quality control, batch effect correction, and multi-dimensional analysis, an in-depth characterization of intercellular heterogeneity and subpopulation characteristics was achieved. This data-driven approach, which uses cells as the analysis unit, has been widely applied and recognized in numerous authoritative literatures, and it has strong applicability and explanatory power in exploring cell state transitions, changes in signaling pathways, and potential targets. We have adjusted the text to make the description more accurate.

Concern #2: No validation using independent datasets or wet-lab experiments provided for proposed genes and drugs.

Author response: We fully acknowledge your suggestions that the biological significance of bioinformatics prediction results needs to be further confirmed through in vitro and in vivo experiments. Therefore, in the Results section, we have supplemented the qPCR validation of DUSP6, AP1S2, and PTPRB to enrich the conclusions of this study. For specific details, please refer to Figure 9. Thank you for your valuable suggestions!

Concern #3: Use of CellChatDB.human for mouse scRNA-seq data without justification.

Author response: Thank you for the reviewers' attention to the methodology of this study. In the cell communication analysis of this study, CellChatDB.human was selected as the reference database, mainly because it is superior to CellChatDB.mouse in terms of the completeness of ligand-receptor interaction annotations and pathway coverage, especially in core signaling pathways such as angiogenesis and immune regulation, where the information is more comprehensive. The CellChat development team also pointed out in the original literature that the core ligand-receptor interactions among different species are highly conserved, and cross-species analysis can be achieved through homologous gene mapping. Considering that the number of ligand-receptor entries in CellChatDB.mouse is relatively limited, to avoid missing potential important signals, we performed homologous matching between the human database and mouse genes before the analysis, and only retained the interaction pairs with clear corresponding relationships in the mouse annotations.

Concern #4: Some functional conclusions are speculative and require more accurate interpretations.

Author response: Following your suggestions, we have revised the statements regarding the involvement of VECs in visual perception to make the sentences more speculative rather than absolute. We have further clarified the relevant expressions as preliminary observations and emphasized that the related conclusions of this study are mainly based on single-cell transcriptome data and bioinformatics analysis. Currently, it only suggests that VECs may be involved in related pathways, and their specific physiological functions need to be further verified by subsequent experiments.

Concern #5: Clarification needed on FDR correction and multiple testing adjustments.

Author response: In this study, FDR correction was applied in the procedures of differential expression analysis, enrichment analysis, and related significance statistical processes, with FDR or q-value adopted as the significance criteria. For the AUCell scoring procedure, corresponding treatments were implemented based on its analytical characteristics. The reason for conducting multiple hypothesis test correction lies in the fact that each of our p-values is calculated through a single hypothesis test. We control the α value, that is, the probability of Type I error (the probability of incorrectly rejecting the null hypothesis), and compare it with the p-value. When the p-value is less than the α value, we reject the null hypothesis under the threshold error probability. To manage this error probability, we perform multiple hypothesis test correction to reduce false positives. However, in practice, the results of hypothesis tests fully conform to statistical principles, and not all hypothesis test results necessarily require correction. Forcing multiple corrections may instead lead to over-conservatism (for example, no significant results after correction) and mask real biological signals. Therefore, using the p-value to judge significance is sufficient �PMID: 39923360�. The detailed explanation is as follows:

Differential expression analysis was performed using the limma R package, with the screening criteria set as |logFC| > 0.25 and P value < 0.05. GO and KEGG enrichment analyses were conducted using the clusterProfiler R package, and the significance level was set at P < 0.05. The BEAM branch analysis in Monocle 2 was judged using the q-value (FDR-corrected P value) output by default in the software. The threshold for AUCell gene set activity scoring was automatically determined by the AUCell_exploreThresholds function based on the AUC distribution. Since no significance test was involved, no multiple correction was performed. The determination of significance for CellChat, related network analyses, and correlation analyses all followed the recommendations and default parameter settings of the respective R packages or software. Except for the above-mentioned analyses that did not require hypothesis testing or multiple correction, FDR correction was applied in all other cases involving multiple testing to control the false positive rate. P value < 0.05 was considered statistically significant.

Concern #6: Explanation needed for the threshold defining “glucocorticoid-active” cells.

Author response: In this study, the AUC threshold for GC-active cells was determined based on a comprehensive analysis of the distribution of single-cell gene set enrichment scores using the AUCell tool, and methodologically, it strictly follows the data-driven grouping approach recognized in the current field. The threshold of AUC > 0.13 was automatically recommended by the AUCell_exploreThresholds function according to the AUC distribution of all cells, and was used to distinguish cell populations with high or low gene set activity. We have revised the relevant descriptions in the methods and results sections to enhance readers' understanding of our analytical approach.

130 glucocorticoid-active cells were identified by establishing an ideal threshold, where cell populations with an area AUC value greater than 0.13 were categorized as having high glucocorticoid activity, and those with values less than 0.13 were classified as having low glucocorticoid activity (Fig 3A).

→

130 glucocorticoid-active cells were identified by the defined threshold, where cell populations with an area AUC value greater than 0.13 were categorized as having high glucocorticoid activity, and those with values less than 0.13 were classified as having low glucocorticoid activity (Fig 3A).

Concern #7: Clustering resolution and sensitivity testing not mentioned.

Author response: The resolution parameter of 0.5 was chosen because it yielded clustering results consistent with known retinal cell types and provided clear separation of biologically meaningful populations in our dataset. This value is commonly used in single-cell RNA-seq studies and has been demonstrated in previous literature to deliver reliable results for d

---

## [Decision Letter · Decision Letter 1]

24 Sep 2025

Single-Cell Sequencing Reveals the Response Mechanisms of Vascular Endothelial Cells to Glucocorticoids in Diabetic Retinopathy

PONE-D-25-26641R1

Dear Dr. Zhou,

We’re pleased to inform you that your manuscript has been judged scientifically suitable for publication and will be formally accepted for publication once it meets all outstanding technical requirements.

Kind regards,

Chengming Fan, MD, PhD

Academic Editor

PLOS ONE

Additional Editor Comments (optional):

Reviewer #1:

Reviewer #2:

Reviewer #3:

Reviewers' comments:

Reviewer's Responses to Questions

**Comments to the Author**

Reviewer #1: (No Response)

Reviewer #2: All comments have been addressed

Reviewer #3: All comments have been addressed

2. Is the manuscript technically sound, and do the data support the conclusions?

Reviewer #1: (No Response)

Reviewer #2: Yes

Reviewer #3: Yes

3. Has the statistical analysis been performed appropriately and rigorously?

Reviewer #1: (No Response)

Reviewer #2: I Don't Know

Reviewer #3: Yes

4. Have the authors made all data underlying the findings in their manuscript fully available?

Reviewer #1: (No Response)

Reviewer #2: Yes

Reviewer #3: Yes

5. Is the manuscript presented in an intelligible fashion and written in standard English?

Reviewer #1: Yes

Reviewer #2: Yes

Reviewer #3: Yes

Reviewer #1: The manuscript now presents a coherent and convincing story. It leverages state-of-the-art single-cell sequencing technology to elucidate a pathophysiologically relevant question with potential clinical ramifications. The data is robust, the analysis is sound, and the conclusions are well-supported.I recommend that this manuscript be accepted for publication.

Reviewer #2: Thank you for addressing my comments and for the revisions made to the manuscript. I am satisfied with the responses and the changes implemented, and I consider the manuscript suitable for publication in its current form.

Reviewer #3: (No Response)

**Do you want your identity to be public for this peer review?** For information about this choice, including consent withdrawal, please see our Privacy Policy

Reviewer #1: **Yes: ** saijun zhou

Reviewer #2: No

Reviewer #3: No

---

## [Editor Report · Acceptance letter]

PONE-D-25-26641R1

PLOS ONE

Dear Dr. Zhou,

I'm pleased to inform you that your manuscript has been deemed suitable for publication in PLOS ONE. Congratulations! Your manuscript is now being handed over to our production team.

Kind regards,

on behalf of

Dr. Chengming Fan

Academic Editor

PLOS ONE